# Optimization and Molecular Mechanism of Novel α-Glucosidase Inhibitory Peptides Derived from Camellia Seed Cake through Enzymatic Hydrolysis

**DOI:** 10.3390/foods12020393

**Published:** 2023-01-13

**Authors:** Yuanping Zhang, Fenghua Wu, Zhiping He, Xuezhi Fang, Xingquan Liu

**Affiliations:** 1College of Food and Health, Zhejiang Agriculture and Forestry University, Hangzhou 311300, China; 2Research Institute of Subtropical Forestry, Chinese Academy of Forestry, Hangzhou 311400, China

**Keywords:** camellia seed cake protein (CSCP), *α-glucosidase*, Inhibition kinetic, molecular docking

## Abstract

In recent years, food-derived hypoglycemic peptides have received a lot of attention in the study of active peptides, but their anti-diabetic mechanism of action is not yet clear. In this study, camellia seed cake protein (CSCP) was used to prepare active peptides with α-glucosidase inhibition. The optimization of the preparation of camellia seed cake protein hydrolyzed peptides (CSCPH) was conducted via response surface methodology (RSM) using a protamex with *α-glucosidase* inhibition as an indicator. The optimal hydrolysis conditions were pH 7.11, 4300 U/g enzyme concentration, 50 °C hydrolysis temperature, and 3.95 h hydrolysis time. Under these conditions, the *α-glucosidase* inhibition rate of CSCPH was 58.70% (IC_50_ 8.442 ± 0.33 mg/mL). The peptides with high *α-glucosidase inhibitory activity* were isolated from CSCPH by ultrafiltration and Sephadex G25. Leu-Leu-Val-Leu-Tyr-Tyr-Glu-Tyr (LLVLYYEY) and Leu-Leu-Leu-Leu-Pro-Ser-Tyr-Ser-Glu-Phe (LLLLPSYSEF) were identified and synthesized for the first time by Liquid chromatography electrospray ionisation tandem mass spectrometry (LC-ESI-MS/MS) analysis and virtual screening with IC_50_ values of 0.33 and 1.11 mM, respectively. Lineweaver-Burk analysis and molecular docking demonstrated that LLVLYYEY was a non-competitive inhibitor of *α-glucosidase*, whereas LLLLPSYSEF inhibited *α-glucosidase*, which displayed a mixed inhibition mechanism. The study suggests the possibility of using peptides from Camellia seed cake as hypoglycaemic compounds for the prevention and treatment of diabetes.

## 1. Introduction

Diabetes is becoming more prevalent worldwide and by 2015, the total number of diabetics is expected to reach 700 million [1]. The International Diabetes Federation (IDF) reports that the digestion of food produces excess absorbable monosaccharides that may lead to increased blood glucose levels, leading to hyperglycemia [2,3]. Once a person is diagnosed with diabetes, they will need to take long-term medication to keep their blood glucose levels under control. Inhibiting the activity of carbohydrate digestive enzymes to delay carbohydrate digestion is an efficient method to avoid diabetes or regulate postprandial blood glucose [4]. The best treatment for diabetes mellitus is to maintain appropriate blood glucose levels after meals [5]. As a result, *α-glucosidase* inhibitors, which catalyze the separation of glucose from disaccharides, are efficient in delaying glucose absorption. The inhibition of *α-glucosidase* activity is thought to be a useful method for diabetes management. In light of this, an increasing number of researchers are concentrating their efforts on discovering more efficient *α-glucosidase* inhibitors in natural substances [6].

Camellia oleifera Abel is a Camellia genus in the Camelliaceae family. It has been cultivated for over 2000 years and is primarily found in the hilly highlands of southern China, particularly in the Jiangxi Gan’s southern region [7]. China boasts a diverse range of Camellia trees as well as abundant resources. Camellia oil cake is a by-product of oil tea manufacturing made from the seed residue of Camellia oleifera Abel. Traditionally, it has been used primarily as animal feed or as a source of heat, and its biological potential has not been properly exploited [8]. Active ingredients in Camellia Oil Cake such as sasanquasaponin (SQS), flavonoids, and tannins have been proven to have anti-inflammatory, antibacterial, antioxidant, and anticancer properties [9]. Researchers have extracted peptides that inhibit *α-glucosidase* from camellia seed cake and identified GHSLESIK, GLTSLDRYK, and SPGYYDGR [10], and LPLLR was identified from walnuts [11], as well as other peptides that inhibit *α-glucosidase* from cereals. The role of CSCPH in *α-glucosidase* inhibition, as well as the relationship between peptide structure and inhibition effect, have yet to be thoroughly investigated.

Response surface methodology (RSM) is a statistical method that is often used to explore the relationship between several factors and to find the optimal process parameters through the analysis of regression equations [12]. Determine the characteristics of the Box and Behnken test regions using the Box-Behnken design (BBD), a three-level fractional factorial design. The design is a two-level factorial design for combining incomplete groups of zones, wherein in each module, a certain number of factors are placed through all combinations of the design, while other factors remain at the central level [13]. The BBD-based response surface design has been successfully applied to optimize the hydrolysis conditions for the preparation of α-glucosidase inhibiting peptides from plant-based proteins [14,15].

In earlier studies, peptides with antidiabetic activity were isolated from plant-based proteins [16,17]; however, little is known about the literature on the *α-glucosidase inhibitory* potential of the Camellia seed Cake (CSC) protein and its hydrolyzed peptides. Therefore, the aim of this study was to prepare peptides with α-glucosidase inhibitory peptides by enzymatic process optimization, followed by the purification of crude peptide solutions using sequential chromatographic techniques such as ultrafiltration (UF) membranes with different molecular weight cut-off values (MWCO), and Sephadex gel chromatography. LC-ESI-MS/MS was used to identify the amino acid sequences of the α-glucosidase inhibitor peptides. Finally, the mechanism of *α-glucosidase* inhibition by the peptides was investigated using inhibition kinetics analysis and molecular docking. Therefore, the focus of this study was to optimize the preparation conditions of CSCPH, to identify and screen the *α-glucosidase inhibitory* peptides, and to investigate its mechanism of inhibition of *α-glucosidase*.

## 2. Materials and Methods

### 2.1. Materials

Camellia seeds were sourced from Quzhou, Zhejiang Province, China. Flavourzyme (60 U/mg), alkaline protease (200 U/mg), and Sephadex (G25) were purchased from Beijing Solarbio Biotechnology Co., Ltd. (Beijing, China). *α-glucosidase* (200 U/mg), trypsin (250 U/mg), and protamex (120 U/mg) were from Shanghai Yuanye Biotechnology Co., Ltd. (Shanghai, China). Dithiothreitol (DTT), o-phenylaldehyde (OPA), and sodium dodecyl sulfate (SDS) were purchased from Aladdin Reagents (Shanghai) Co., Ltd. (Shanghai, China), and serine from Shanghai Baiyan Bio-Technology Co., Ltd. (Shanghai, China). All other chemicals and reagents are of analytical grade.

### 2.2. Preparation of CSCP

The treated defatted desaponin powder was weighed and added to distilled water in a large beaker at a ratio of 1:20 (*w*/*v*), and the pH of the solution was adjusted to 10 with 1 M NaOH and placed in a 50 °C water bath for 2 h. The water bath was stirred with mechanical stirring. The alkaline extract was centrifuged at 3800 r/min for 15 min, and the supernatant was collected and adjusted to pH 4.5 with 1 M HCl and left for 1 h. The supernatant was discarded after centrifugation at 3800 r/min for 15 min. The precipitate was re-dissolved with a small amount of distilled water and the pH of the solution was adjusted to 7 with 1 M NaOH. The solution was then lyophilized and stored at −20 °C.

### 2.3. Preparation of CSCPH

According to the substrate concentration of 1% *w*/*v* (g/mL), a certain amount of CSCP was weighed and distilled water was added to make a CSCP solution. The solution was denaturated in a 95 °C water bath for 10 min and then cooled to room temperature. Enzymatic digestion was for 4 h at optimum pH and temperature. After enzymatic hydrolysis, the enzyme was killed in a 95 °C water bath for 10 min, then cooled to room temperature, centrifuged at 4 °C at 8000 r/min for 30 min, and the supernatant was lyophilized and stored at −20 °C.

### 2.4. Optimization of Enzymatic Parameters via RSM

In this study, the enzymatic hydrolysis conditions of CSCP were optimized based on the Box-Behnken design of RSM. The experimental design used temperature (A), pH (B), hydrolysis time (C), and protease concentration (D) as independent variables, while the selected response variable (Y) was the inhibition rate of α-glucosidase. The settings of the factors and levels are shown in Appendix A.

### 2.5. Fractionation and Purification of α-Glucosidase Inhibitory Peptides

#### 2.5.1. Ultrafiltration

After the identification of the CSCPH with the highest α-glucosidase inhibitory activity, further fractionation was carried out using an ultrafiltration unit (Mini Pellicon) (Millipore, Billerica, MA, USA) and three MWCO (molecular weight cut-off) membranes of 10, 3, and 1 kDa. Four molecular weight fractions were obtained, which were >10 kDa, 3–10 kDa, 1–3 kDa, and <1 kDa. Then, these fractions were lyophilized and the *α-glucosidase inhibitory activity* was measured. The fraction with the highest α-glucosidase inhibition potential was selected for the next step, lyophilized, and stored at −20 °C for use in subsequent experiments.

#### 2.5.2. Sephadex G25 Chromatography

The fractionation with the greatest *α-glucosidase inhibitory activity*, which was prepared by ultrafiltration as chosen in 2.5.1, was applied onto a Sephadex G25 column (1.6 cm × 80 cm) and balanced with ultra-pure water at a flow rate of 0.6 mL/min. CSCPH-II was filtered by a 0.22 μm filter. Subsequently, 5 mL (4 mg/mL) of the sample solution was loaded onto a well-balanced Sephadex G25 column, and the different fractions were collected at a flow rate of 0.6 mL/min. Fractions were collected using an automatic partial collector (3 min/tube). The absorbance of the sample was measured at 280 nm.

### 2.6. Assaying the Inhibitory Action of α-Glucosidase In Vitro

The α-glucosidase inhibition activity was performed as described in the literature [18]. In a 96-well enzyme plate, 40 μL of PBS buffer solution (pH 6.86, 0.1 M), 40 μL of sample solution, and 80 μL of α-glucosidase solution (0.2 U/mL) were added, mixed well, and reacted at 37 °C for 15 min. Next, 80 μL of 2.5 mM p-nitrophenyl-α-D-glucopyranoside (pNPG, 0.1 M, pH 6.8 in PBS) was added, mixed well, and reacted at 37 °C for 20 min. Finally, 150 μL of 0.2 M Na_2_CO_3_ solution was added to stop the reaction, which was then measured for absorbance values at 405 nm *A*_1_. Data were collected for three parallel experiments.
(1)$$\alpha \text{-}\mathrm{glucosidase}\;\mathrm{inhibitory}\;\mathrm{activity}\;\left(\%\right)\;=\;\frac{A_{0}-{\;(A}_{\;1}-{\;A}_{2})}{{\;A}_{0}}\;\times \;100\%$$
where *A*_0_ is the absorbance of an equal amount of PBS buffer in place of the sample; *A*_2_ is the absorbance of an equal amount of PBS buffer in place of the *α-glucosidase* solution and pNPG solution.

### 2.7. O-Phthaladehyde (OPA) Assay of Hydrolysis Degree (DH) of CSCPH

In this study, the hydrolysis degree of CSCPH was determined according to the method proposed by Nielsen et al. [19]. In brief, 400 μL of CSCPH and 3 mL of prepared OPA were added sequentially to the test tube. The mixture was rapidly mixed and allowed to stand for 2 min at room temperature before the absorbance value was measured at 340 nm. All tests were repeated three times. *DH* is calculated by Equation (2):(2)$$\mathrm{DH}\;=\;\frac{h}{h_{\mathrm{tot}}}\;\times \;100\%$$
where *h* represents the total number of hydrolyzed bonds (*h* = (serine NH_2_ − β)/αmeqV/g protein) and *h_tot_* represents the total number of peptide bonds found in the CSCP substrate (7.8 Mequiv/g). The values of α and β depend on the amino acid composition of the protein used as raw material. *Serine NH_2_* was calculated according to Equation (3):(3)$${\mathrm{Serine}\text{-}\mathrm{NH}}_{2}\;=\;\frac{{(\mathrm{OD}\;}_{\mathrm{sample}}\;-\;{\mathrm{OD}}_{\mathrm{blank}})}{{(\mathrm{OD}}_{\mathrm{standard}}\;-\;{\mathrm{OD}}_{\mathrm{blank}})}\;\times \;\frac{0.9516\;\mathrm{meqv}/L\times 0.1\times 100}{X\times P}$$
where *OD_sample_* is the absorbance of the sample solution, *OD_blank_* is the absorbance of the equal volume of water instead of the sample, *OD_standard_* is the absorbance of the equal volume of serine solution at 340 nm, *X* is the number of g samples, *P* is the percentage of protein in CSCP (79.05%), and the total sample volume is 0.1 L.

### 2.8. Examination of Amino Acids

The composition of amino acids was determined by high-performance liquid chromatography (HPLC) derived from the PITC column. The mobile phase was 0.1 M sodium acetate buffer/acetonitrile (97:3, *v*/*v*) and acetonitrile/water (4:1, *v*/*v*). Inject 10 µL sample solution directly into the C18 Inertsil ODS-SP column (4.6 mm × 250 mm, 5 µm), the flow rate is 1.0 mL/min. The HPLC system (LC-20AT, Shimadzu, Kyoto, JPN) was used to monitor the solution at 254 nm and 40 °C.

### 2.9. Identification of the α-Glucosidase Inhibitory Peptides

The peptide sequence of CSCPH-II-4 was identified by Bio-Tech Pack Technology Co. Ltd. (Beijing, China). In brief, the fractions CSCPH-II-4 were sequenced using a Nano UPLC-MS/MS system. Experiments were performed on an Easy-nLC1200 system and a Q Exactive™ Hybrid Quadrupole-Orbitrap™ mass spectrometer (Thermo Fisher Scientific, Waltham, MA, USA) with an ESI nanospray source. Raw MS files were analyzed and searched for these sequences to match parent proteins based on sample species using Peaks Studio, and the database used for searching was the camellia proteome reference database from UniProt.

### 2.10. Virtual Screening

Schrödinger Maestro is a useful molecular simulation software, which was used in this study to screen promising peptides [20]. The peptide structures were constructed by ChemDraw. The Glide module in the Schrödinger Maestro software was used for virtual screening, and the Protein Preparation Wizard module was used to process the target protein (PDB ID:2QMJ), remove the water of crystallization, add missing hydrogen atoms, and repair missing bond information. Finally, the protein underwent energy minimization and geometry optimization. The receptor was minimized using the OPLS3e force field, and all peptide molecules were prepared according to the default settings of the LigPre module. For screening in the Glide module, the prepared receptor protein files were imported to specify the appropriate position in the receptor grid generation. Glide extra precision (XP) represents a single, coherent approach, where the sampling algorithm and the scoring function have been optimized concurrently [21]; using glide/XP scoring, peptides with high scores were screened for in vitro activity validation using the docking template. The screening process used acarbose as a positive control to determine the binding pattern of the original ligand to the active site.

### 2.11. Synthesis of α-Glucosidase Inhibitory Peptides

Based on the results of the LC-ESI-MS/MS analysis and virtual screening, five peptides: Leu-Leu-Val-Leu-Tyr-Tyr-Glu-Tyr (LLVLYYEY), Leu-Leu-Leu-Leu-Pro-Ser-Tyr-Ser-Glu-Phe (LLLLPSYSEF), Leu-Cys-Asp-Gln-Cys-Pro-Pro-His-Ala (LCDQCPPHA), Ala-Thr-Asn-Pro-Pro-Cys-Cys-Gln-Pro (ATNPPCCQP), and Lys-Asp-Asp-Phe-Val-Glu-Lys-Arg (KDDFVEKR) were synthesized by the GenScript Co., Ltd. (Nanjing, China). In brief, the peptides were detected by RP-HPLC with purity above 95% (*w*/*w*). Mobile phase A, 0.065% TFA; mobile phase B, 0.05% TFA acetonitrile solution; flow rate, 1 mL/min; column, Inertsil ODS-3, 4.6 mm × 250 mm, detected at 220 nm. Finally, the purified peptides were identified by ESI-MS spectroscopy.

### 2.12. Molecular Docking

In this study, the interactions between the peptide and the target protein were estimated using molecular docking [22]. The X-ray crystal structure of *α-glucosidase* (PDB ID 2QMJ) was obtained from the Protein Data Bank (www.rcsb.org, accessed on 22 July 2022) by Autodock Tool 1.5.6 Software; the downloaded receptor protein was dehydrated and hydrogenated. 2QMJ had active centers of X: −20.83, Y: −6.71, and Z: −5.25. The 3D structure of the peptide was constructed and energy was minimized using ChemBio 3D software. Semi-flexible docking was performed using the Vina module and the number of generated docked conformations was set to 20. Based on the least energy score, the optimum binding conformation for the peptide and α-glucosidase was chosen among all docking results. Visual analysis was performed by PyMOL.

### 2.13. Mechanism of α-Glucosidase Inhibitions

With minor modifications, a recent study determined the kinetic parameters of *α-glucosidase* inhibition [23]. In brief, different concentrations of peptide solutions were prepared and mixed with 6 μg/mL *of α-glucosidase* and incubated at various concentrations of pNPG (0.8, 1, 2, 3, 4 m M), as described in 2.6. The Linerweave-Burk double inverse method was used to make a graph, with 1/[*S*] as the horizontal coordinate and 1/V as the vertical coordinate, to plot straight lines at different concentrations to determine the type of inhibition and calculate the relevant parameters, and the double inverse equation is shown in Equation (4). Secondary graphing is calculated as shown in Equations (5) and (6).
(4)$$\frac{1}{V}\;=\;\frac{K_{m}+[S]}{V_{\max}\times [S]}$$
(5)$$\mathrm{Slope}\;=\;\frac{K_{m}}{V_{\max}\frac{[I]\times K_{m}}{K_{\mathrm{ic}}{\times V}_{\max}}}$$
(6)$$\frac{1}{V}\;=\;\frac{K_{m}}{{[S]\times V}_{\max}}\;\times \;(1+\frac{\left[I\right]}{K_{\mathrm{ic}}})\;+\;\frac{1}{V_{\max}}\;\times \;(1+\frac{\left[I\right]}{K_{\mathrm{iu}}})$$

Which *V* is the reaction rate. [*S*] and [*I*] are the concentrations of substrate and inhibitors, respectively. *K_m_* is the Michaelis-Menten constant, *V_max_* is the maximum reaction rate, *K_ic_* is the competitive inhibition constant, and *K_iu_* is the uncompetitive inhibition constant.

### 2.14. Simulated Gastrointestinal Digestion In Vitro

The method of simulated gastrointestinal digestion (SGID) follows the previous method with slight modifications [24]. Simulated gastric fluid (SGFs) and simulated intestinal fluid (SIFs) are both purchased from Shanghai Yuanye Biotechnology Co., Ltd., as previously mentioned. In brief, the synthesized peptide LLVLYYEY and LLLLPSYSEF was pre-dissolved to 4 mg/mL in pure water. LLVLYYEY and LLLLPSYSEF aqueous solution was mixed with SGFs (1:1, *v*/*v*) and digested at 37 °C for 120 min. SIFs (1:1, *v*/*v*) was then mixed and the mixture was incubated for 120 min. After 120 min of incubation, the pH was adjusted to 7 by NaOH (1M) to stop the simulated gastric digestion (SGD) phase; after 240 min of incubation the simulated intestinal digestion (SID) phase was stopped by a boiling water bath for 10 min.

### 2.15. Statistical Analysis

The results were analyzed using SPSS version 26. Statistical analysis was performed using one-way ANOVA and Duncan’s multiple-range test. Differences were considered significant at *p* < 0.05. All data are reported as mean ± SD. All experiments were repeated in triplicate.

## 3. Results and Discussion

### 3.1. Protease Screening

In order to investigate whether different peptides are effective, we used different proteases to treat the same protein substrate based on their specific enzymatic cleavage sites and obtained peptides that may have different structures and biological activities [25]. As a result, it is necessary to screen suitable proteases for peptides with strong hypoglycemic activity. Four typical endoproteases were employed in this investigation to hydrolyze CSCP. Since *α-glucosidase* is one of the key enzymes for the fine regulation of insulin function, it is a target for delaying glucose absorption and inhibiting postprandial hyperglycemia. Some *α-glucosidase* inhibitors can inhibit carbohydrate digestion by competitively inhibiting various α-glucosidases in the small intestine; therefore, the rate of *α-glucosidase* inhibition is used as an indicator for selecting the optimal protease. It was discovered that protamex’s hydrolysate was noticeably (*p* < 0.05) more effective than the other aforementioned enzymes (Figure 1). Protamex was therefore selected for this experiment as an expeditious enzyme in producing efficacious bioactive peptides from the CSCP to inhibit *α-glucosidase* activity. The flavourzyme hydrolysate *α-glucosidase* inhibition activity was the lowest, which could be attributed to the special selection of the substrate by different proteases.

### 3.2. Optimization of CSCPH

#### 3.2.1. Preliminary Assessment

During the initial testing, RSM was used to discover the optimum conditions for CSCPH hydrolysis, where the different components were changed one by one to see how they affected the results. The aim was to determine the central point values for these several hydrolysis parameters, including pH (A), hydrolysis time (B), protease concentration (C), and temperature (D). As shown in Figure 2A, the change in pH from 6.0 to 7.0 resulted in an increase in *α-glucosidase inhibitory activity* values and a significant increase in DH values. As the pH exceeded 7.0, the DH value gradually decreased and the *α-glucosidase inhibitory activity* value also decreased slightly. The results indicated that the highest *α-glucosidase inhibitory activity* was obtained for CSCPH in the pH range of 6–8 at the time of enzymatic digestion (Figure 2B). The DH value increased with increasing hydrolysis time, while the *α-glucosidase inhibitory activity* increased at 4 h of hydrolysis. As the enzymatic digestion continued, the *α-glucosidase inhibitory activity* showed a decreasing trend, which was consistent with the results of Gao et al. (2019) [26]. The duration of the follow-up experiment was set at 4 h. This may be due to the fact that the effective fragments with inhibitory activity obtained by hydrolysis at around 4 h were destroyed after successive hydrolysis, and the content of peptides with inhibitory effects on *α-glucosidase* was reduced, resulting in a decrease in the overall inhibitory activity of the hydrolysis product. When the degree of hydrolysis was stabilized, the inhibitory activity also stabilized, indicating that the composition of the peptides in the hydrolysis product remained unchanged, and the content and structure of the peptides with an inhibitory effect on *α-glucosidase* did not change. The inhibitory activity of CSCPH *α-glucosidase* was also affected by enzyme concentration, with the maximum inhibitory activity reaching 4000 U/g and no significant change in inhibitory activity as the enzyme concentration continued to increase (Figure 2C. Similar to the other factors, the DH value increased from 40 ˚C to 50 ˚C and the *α-glucosidase inhibitory activity* increased slightly as well. Above 55 ˚C, the DH value gradually decreased and the *α-glucosidase inhibitory activity* also decreased slightly. The protease is thought to be inhibited at higher temperatures, leading to incomplete enzymatic digestion, which reduces DH and *α-glucosidase inhibitory activity* [27].

#### 3.2.2. Optimization Analysis of RSM

Further optimization of four variables using BBD-based RSM based on an initial assessment of single-factor trials to achieve the best *α-glucosidase* inhibition rate of CSCPH: pH, enzymatic digestion time, enzyme concentration, and temperature. The center point values for the four independent variables had already been fixed at 50 °C, pH 7.0, 4000 U/g, and 240 min in previous studies (Appendix A). The design matrix included 29 experimental runs (Appendix A). The experimental data for the *α-glucosidase inhibitory activity* of CSCPH were analyzed, and the *p*-values and coefficients are shown in (Appendix A). The prediction equation to achieve the maximum *α-glucosidase inhibitory activity* of CSCPH is as follows:Y = + 58.37 + 0.2258A + 1.8B − 0.3058C + 2.83D − 0.42AB + 0.415AC + 0.2775AD − 0.0675BC + 0.1775BD + 0.355CD − 1.95A^2^ − 6.2B^2^ − 2.66C^2^ − 10.62D^2^
(7)
where Y is the *α-glucosidase* inhibition rate (%), and A, B, C, and D is the pH of the enzymatic digestion solution, the enzymatic digestion time, the protease concentration, and the temperature of the enzymatic digestion solution, respectively.

The coefficient of determination (R^2^) was used in conjunction with the lack of fit test and probability (*p*) values to assess model fitness. Table 1 shows the ANOVA findings for the response surface quadratic model explaining *α-glucosidase* inhibitory action, which showed *F*-values of 21.37 and an extremely low *p*-value (*p* < 0.0001), implying that the variables in the model were very significant. Meanwhile, the respective *p*-values of the lack of fit test for the model of *α-glucosidase inhibition activity* were 0.0614, demonstrating that the model was capable of accurately predicting the *α-glucosidase inhibitory activity* for all combinations of the independent variables investigated at the significance level (*p* > 0.05). Furthermore, the purpose of the lack of fit test is to establish whether the experimental data can be adequately described by the model or whether a more complicated model is required. The model of *α-glucosidase inhibitory activity* gave an R^2^ value of 0.9553, suggesting that according to the prediction equation, the model would describe the response variability quite well. Closer R^2^ values approaching 1.00 indicate that the model has a better capacity to predict the response with more accuracy [28]. Meanwhile, the relative adjusted R^2^ values for *α-glucosidase inhibition activity* were 0.9553, suggesting that only 4.47% of the *α-glucosidase inhibitory activity* in this model was not explained. These results demonstrate that the fitting model is a satisfactory mathematical description of the hydrolysis process.

The response surface plots (3D) were generated using a quadratic equation in which the *z*-axis was used to plot the response values of *α-glucosidase* inhibition activity against any other pair of independent variables on the x-axes and y-axes, with the other independent variables held constant at the center point. This statement is supported by the figures. The plots can aid in presenting a clear picture of the interactions that occur between independent variables and their impact on the response variables. Additionally, they can be helpful in identifying the optimal hydrolysate condition that has the highest level of *α-glucosidase inhibition activity*.

The effect of independent variables on the *α-glucosidase inhibitory activity* of the enzymatic digestion product was considered. The enzyme concentration had the greatest effect on the *α-glucosidase inhibitory activity* of the enzymatic digestion product (*p* < 0.0001), followed by pH (*p* = 0.0041), while there was no significant effect of enzymatic digestion time (*p* = 0.5701) and enzymatic digestion temperature (*p* = 0.6741) on the *α-glucosidase* inhibitory activity. To illustrate the calculated interaction of the respective variables on the *α-glucosidase* inhibition activity of the enzymatic digestion products, the interaction of pH, temperature, protease concentration, and time on the *α-glucosidase inhibitory activity* of the enzymatic digestion products is depicted in Figure 3. The degree of the independent variables’ influence on the response values for the *α-glucosidase* inhibition rate is indicated by the steepness of the upper spatial surface of the network plot, and the elliptical eccentricity of the contour plot in the lower part of the image reflects the effect of the independent variables on the *α-glucosidase* inhibition response values. The optimum conditions for the factors influencing enzymatic digestion are represented at the peak of each curved surface. The model produced a desirability value close to 1, which implies that the recommended settings are optimal for obtaining the maximum *α-glucosidase* inhibition activity at CSCPH [19]. The ideal hydrolysis conditions for *α-glucosidase* inhibition were estimated to be 50.23 °C, pH 7.07, 4267 U/g protease concentration, and 3.95 h. These conditions may result in a 58.70% inhibition of *α-glucosidase* activity. Experiments were carried out in triplicate using the expected optimal conditions to confirm the validity of the model. According to the data, the experimental value obtained was a little higher than the predicted value, 59.36 ± 0.69%, with an error value of 0.66%. As the experiments produced results close to the predicted values, the model was considered to anticipate the best-case scenario for the production of the *α-glucosidase* inhibitory peptides.

### 3.3. Separation and Purification

#### 3.3.1. Ultrafiltration

Considering that the molecular weight of protein hydrolysates is so important in the generation of bioactive peptides, ultrafiltration is used to separate the hydrolysate into various fractions with varied molecular weights. Different MWCO (molecular weight cut-off) membranes are often used for this process. This is a method that can be easily scaled up to produce *α-glucosidase* inhibitory peptides on an industrial scale [29]. In this study, CSCPH with high *α-glucosidase inhibitory activity* was selected for further fractionation and ultrafiltration separation into CSCPH-I (<1 kDa), CSCPH-II (1–3 kDa), CSCPH-III (3–10 kDa), and CSCPH-IV (>10 kDa). Table 2 shows the *α-glucosidase inhibitory activity*, which is clearly molecular weight dependent. CSCPH-II had the strongest *α-glucosidase inhibitory activity* (IC_50_ 3.896 ± 0.148 mg/mL), while CSCPH-III had the least inhibitory effect on *α-glucosidase* (IC_50_ 62.44 ± 0.965 mg/mL). In general, peptides with lower molecular weights have higher potential inhibitory activity [30]; however, CSCPH-I was identified to have the lowest *α-glucosidase* inhibitory effect, possibly due to the abundance of free amino acids and salt ions in it, resulting in a reduced positive effect. These findings suggest that reducing the molecular weight of the hydrolysate fraction may increase the maximum *α-glucosidase* inhibition of the peptide. It is also worth noting that the *α-glucosidase* inhibition of peptides of different molecular weights is influenced by the amino acid composition and sequence.

#### 3.3.2. Separation of *α-Glucosidase* Inhibitory Peptides by Gel Filtration Chromatography

Sephadex G25 (1.6 × 80 cm) was used to separate CSCPH-II. Four fractions were collected based on varied elution times: CSCPH-II-1, CSCPH-II-2, CSCPH-II-3, and CSCPH-II-4 (Figure 4A), with the inhibition activity of *α-glucosidase* being IC_50_ 13.641 ± 0.37 mg/mL, 11.376 ± 0.122 mg/mL, 7.021 ± 0.096 mg/mL, and 2.033 ± 0.093 mg/mL, respectively (Figure 4B). The effect of CSCPH-II-4 was more pronounced, implying that CSCPH-II-4 had greater *α-glucosidase* inhibitory action. This could be related to the fact that CSCPH-II-4 has a lower average molecular weight than the other three components. Gel filtration chromatography is often used to separate water-soluble macromolecules, and it has been widely used for the separation and desalting of mixed components with a good effect for the separation and purification of peptides [31]. The *α-glucosidase* inhibition of CSCPH-II-4 was higher than that of other fractions, possibly because the average molecular weight of CSCPH-II-4 was lower. Yao et al. (2016) found that after molecular weight grading, peptide mixtures with low molecular weight showed stronger effects [29], which agreed with the findings of this investigation.

### 3.4. Amino Acid Analysis

As shown in Table 3, the proportions of amino acids varied considerably between the different stages of purification. The essential amino acid (EAA) ratio of purified CSCPH-II-4 was 37.50 ± 0.41%, which was significantly higher than that of fresh CSCP (31.51 ± 0.13%). In addition, branched-chain amino acids (BCAAs), including Leu and Val, showed the same upward trend as EAA. In addition, the levels of His and Arg were significantly higher than those of their parents. The proportions of specific amino acids (Glu and Pro) in CSCPH-II-4 were 10.42 ± 0.08% and 13.19 ± 0.13%, respectively. Amino acids in the diet have different degrees of influence on the body’s metabolism. EAA is not usually created by our body, however, it is necessary for our metabolism. Therefore, it is often used to assess the nutritional value of food proteins [32]. Moreover, BCAA is another amino acid of interest, as it has been shown to have beneficial effects on exercise, sports nutrition, and improved liver function [29,33]. Thus, the CSCPH-II-4 composition recovered from Sephadex G25 is rich in EAA and BCAA, particularly Pro and Leu, suggesting that it has great potential in creating nutrients for humans, particularly hyperglycaemic patients.

### 3.5. Screening for Peptides with α-Glucosidase Inhibitory Activity

To identify peptides with *α-glucosidase inhibitory activity*, the sequence peptides in CSCPH-II-4 were identified using LC-ESI-MS/MS. Peptide sequence resolution of the mass spectrometry raw files using the software PEAKS Studio (8.5) ‘de novo score’ resulted in the identification of 469 peptides. Molecular docking is now becoming a common tool for fast and effective virtual drug screening and design. Peptides with higher scores were further validated for in vitro activity through docking performed by the Glide/XP scoring function [34] for de novo scores > 90 (Appendix A).

The virtual screening method used 2QMJ as a representative protein structure for *α-glucosidase* with high accuracy (resolution of 1.90 Å) and used acarbose as the initial ligand to ascertain the ligand’s method of binding to the *α-glucosidase* active site (Figure 5 A,B) to construct the *α-glucosidase* active site. *α-glucosidase* active site amino acid residues ASP203, THR205, ARG526, HIS600, ASP542, and ASP327 play an important role in the formation of the active pouch and stabilization of the original ligand (Figure 5B). This is consistent with the findings of Liu et al. (2021), who identified LDLQR, AGGFR, and LDNFR from WGPs with *α-glucosidase inhibitory activity* [2]. These amino acids form multiple hydrogen bonds at the active site and interact with acarbose, setting the stage to screen possible inhibitors of *α-glucosidase*.

The first five promising peptides were screened and scored for in vitro activity validation (Table 4), and the MS/MS spectra and structures of LLVLYYEY and LLLLPSYSEF revealed that they had high *α-glucosidase inhibitory activity*, as shown in Figure 6A,B. The lower the energy required for the peptide to bind to α-glucosidase, the easier it will be to bind. LLVLYYEY and LLLLPSYSEF had predicted the binding energies of −9.355 and −9.060 kcal/mol, respectively, with binding energies below −6 kcal/mol, indicating the theoretical *α-glucosidase inhibitory activity* of these two peptides [2]. Similar binding energy scores for *α-glucosidase* inhibitory peptides have also been reported by Mohammed et al. (2018), who utilized in silico-*designed* peptide sequences with binding energy scores ranging from −6.3 to −8.7 kcal/mol [35]. The results of the screening were consistent with the results of amino acid analysis, and according to a previous report on the relationship between structure and activity of peptides, the inhibitory activity of *α-glucosidase* was strongly influenced by two hydrophobic amino acids: Pro and Leu [36]. For instance, the walnut peptide LFLLR was shown to be a prospective *α-glucosidase* inhibitor [11], and this was further validated by the fact that both peptides screened in this study contained high levels of Leu.

### 3.6. Mechanism of α-Glucosidase Inhibition

#### 3.6.1. Kinetics of *α-Glucosidase* Inhibition

To further investigate the potential mechanism of inhibition of *α-glucosidase* by LLVLYYEY and LLLLPSYSEF, the kinetics of inhibition of *α-glucosidase* were determined using the Lineweaver-Burk plot analysis. Figure 7A shows that as the concentration of LLVLYYEY increased, *V_max_* gradually decreased and the double inverse plotting line nearly intersected the *X*-axis, indicating that the *K_m_* value remained unchanged. This indicates that the type of inhibition of *α-glucosidase* by LLVLYYEY is non-competitive, that is, the peptide is not structurally similar to the substrate and does not occupy the active center of the enzyme with the substrate, but rather inhibits the enzyme activity by binding to an essential group other than the active center. Figure 7B shows that LLLLPSYSEF inhibited *α-glucosidase* in a mixed manner, including both competitive and non-competitive inhibition, as both *K_m_* and *V_max_* values increased with the increasing substrate concentration. In addition, we performed a secondary plot of the peptide concentration I using slope and 1/*V_max_* (min/∆OD), respectively, and the fits all yielded a simple straight line, indicating a single inhibition site or a single inhibition-like site on *α-glucosidase* [37]. As shown in Figure 4E, the non-competition constant *K_iu_* for LLVLYYEY was 0.547 mg/mL, as previously reported by Wu et al. (2020) [38], indicating that LLVLYYEY has a good affinity for α-glucosidase. As shown in Figure 4D, the competitiveness of LLLLPSYSEF was 0.9125 mg/mL in the competitive inhibition (*K_ic_*) and 3.76 mg/mL in the non-competitive inhibition (*K_iu_*) (Figure 4F). For LLLLPSYSEF the *K_ic_* value was less than the *K_iu_* value, indicating that LLLLPSYSEF binds with greater affinity to the free enzyme than to the enzyme-substrate (ES) complex. The inhibition mechanism could indicate that LLLLPSFSEF first binds to *α-glucosidase* to form a competitively inhibited enzyme inhibitor (EI) complex, and due to the non-competitive inhibition of LLLLPSFSEF, it can further bind to ES complex to form an enzyme-substrate inhibitor (ESI) complex. These results suggest that LLVLYYEY is a non-competitive *α-glucosidase* inhibitor, whereas LLLLPSYSEF is a mixed *α-glucosidase* inhibitor.

#### 3.6.2. Molecular Docking

Molecular docking is a bioinformatics approach that assesses potential interactions between receptors and ligands by studying their sites of action and key residues [39]. Sites, docking energies, and key residues are used to assess the potential interactions between receptors and ligands. Molecular docking was applied to simulate the binding pattern of LLVLYYEY and LLLLPSYSEF to α-glucosidase, the binding mode of the identified Camellia seed protein peptide to *α-glucosidase* (Figure 8). As noted in Figure 8A,B, both new peptides ‘fit’ into the internal cavity of the *α-glucosidase* enzyme and may create hydrogen bonds with a variety of residues of amino acids inside the internal cavity. For instance, LLVLYYEY forms seven hydrogen bonds with active sites ARG730, GLY732, ARG653, and GLU661 with bond lengths ranging from 1.8 to 2.6 Å, with an average bond length of 2.21 Å. LLLLPSYSEF forms five hydrogen bonds with active sites ASP203, THR205, TYR605, and GLN603 with bond lengths ranging from 2.0 to 2.7 Å, with an average bond length of 2.44 Å. Similar findings were made by Abraham et al. (2014), who discovered that WVYY (seven bonds) had greater ACE inhibitory action than WYT (four bonds) [40]. It is also important to note that the inhibitory capacity appeared to be directly related to the number of hydrogen bonds formed. This indicates that hydrogen bonding is the main force between LLVLYYEY, LLLLPSYSEF, and *α-glucosidase* [41]. The *α-glucosidase* crystal active sites used in this study included ASP203, THR205, ASP542, and ARG526 [42]; however, none of these residues of amino acids reacted with LLVLYYEY. This points to an encounter that is not competitive. However, two bonds were created with ASP203 and one with THR205 out of the five hydrogen bonds that were generated between LLLLPSYSEF and α-glucosidase, suggesting a competitive interaction between LLLLPSYSEF and α-glucosidase. This is in line with the findings of enzyme inhibition kinetic studies.

### 3.7. α-Glucosidase Inhibitory Peptides Simulated Gastrointestinal Digestion In Vitro

Simulated physiological digestion is a very useful tool for evaluating the stability of bioactive peptides against digestive enzymes [43]. The results in Figure 9 show that there was essentially no loss of *α-glucosidase inhibitory activity* after simulated gastric digestion for LLVLYYEY and LLLLPSYSEF, but after SGID they both demonstrated a significant loss of activity. Similar results indicated that YPVEPF was only slightly degraded during SGD, whereas after SGID, YPVEPF was substantially degraded [44]. This reflects the digestive process of food in vivo and suggests that the two novel *α-glucosidase inhibitory* peptides are highly resistant to SGD and still have *α-glucosidase inhibitory activity* after SGID.

## 4. Conclusions

CSC is a by-product of the production process of Camellia seed oil, and the use of these by-products to produce high-value-added functional food ingredients is a timely and popular area of research in this field. In this study, we developed a model to optimize CSCP for the preparation of *α-glucosidase* inhibiting active peptides. CSCPH was isolated and purified using ultrafiltration and Sephadex G25 to obtain CSCPH-II-4. Two bioactive peptides, LLVLYYEY and LLLLPSYSEF, were identified from CSCPH-II-4 for the first time by LC-ESI-MS/MS and virtual screening. Inhibition kinetics and molecular docking results indicated that LLVLYYEY was a non-competitive inhibitor, whereas LLLLPSYSEF was a mixed inhibitor. These peptides have not been found in any natural sources up to this point. Future research will focus on these peptides’ antidiabetic efficacy and toxicity features, paving the way for the creation of these peptides as dietary supplements or anti-diabetic medications.

## Figures and Tables

**Figure 1 foods-12-00393-f001:**
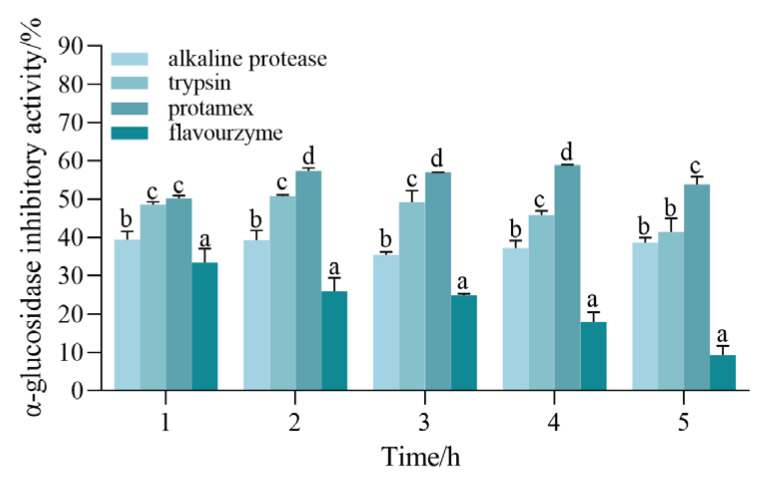
Comparison of *α-glucosidase inhibitory activity* during hydrolysis with four proteases. All values are expressed as the mean ± standard deviation of *α-glucosidase* inhibition. Bias and testing are performed in triplicate. Different superscript letters in the same group indicate that they are significantly different (*p* < 0.05).

**Figure 2 foods-12-00393-f002:**
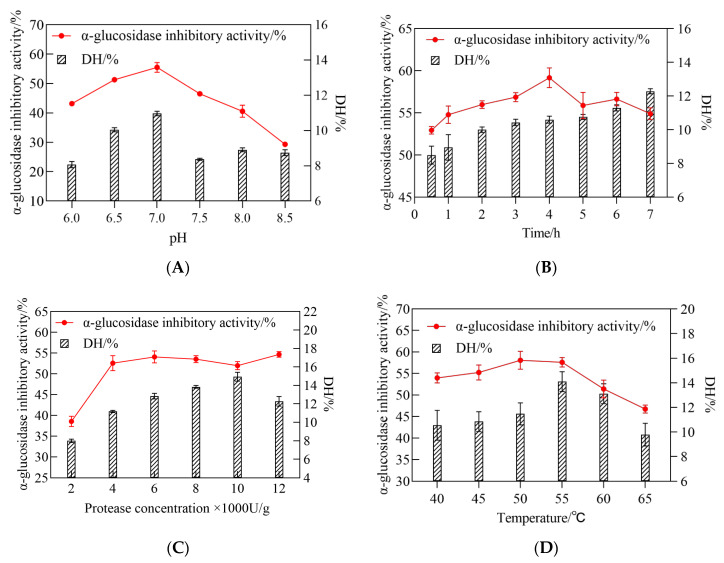
Influence of pH (**A**), hydrolysis time (**B**), protease concentration (**C**), and temperature (**D**), upon *α-glucosidase inhibitory activity* and DH of the hydrolysate. All values are expressed as the mean ± standard deviation of *α-glucosidase* inhibition. Bias and testing are performed in triplicate.

**Figure 3 foods-12-00393-f003:**
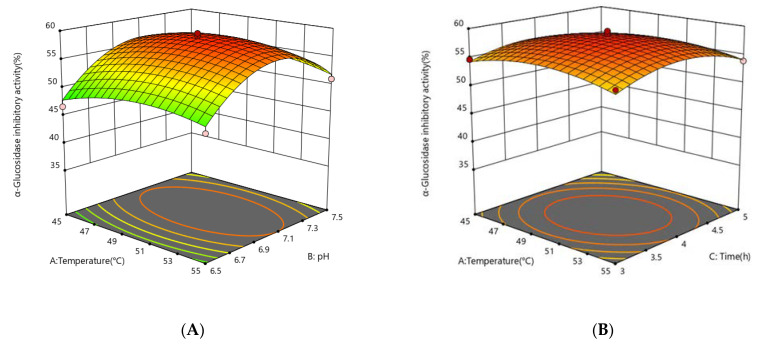
Response surface showing the impact of interactions between independent variables on the activity of *α-glucosidase* inhibition. (**A**) the interaction between temperature and pH; (**B**) the interaction between temperature and time; (**C**) the interaction between temperature and protease concentration; (**D**) the interaction between protease concentration and pH; (**E**) the interaction between protease concentration and time; (**F**) the interaction between protease pH and time. The mean ± standard deviation of *α-glucosidase* inhibition is used to express all values. Bias and testing are performed in triplicate.

**Figure 4 foods-12-00393-f004:**
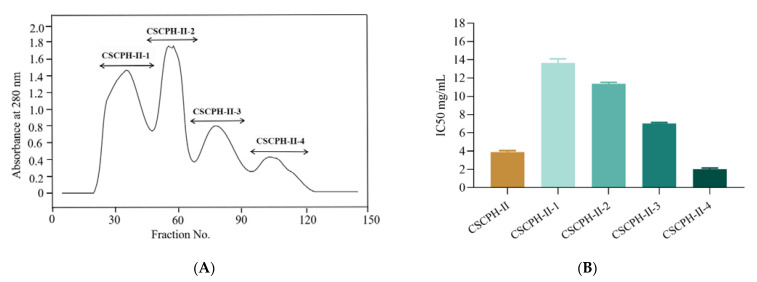
Elution profile of CSCPH-II purification by Sephadex G25 (**A**); *α-glucosidase inhibitory activity* (IC_50_ mg/mL) of CSCPH-II-1, CSCPH-II-2, CSCPH-II-3, and CSCPH-II-4 fractions (**B**). All values are expressed as the mean ± standard deviation of *α-glucosidase* inhibition. Bias and testing are performed in triplicate.

**Figure 5 foods-12-00393-f005:**
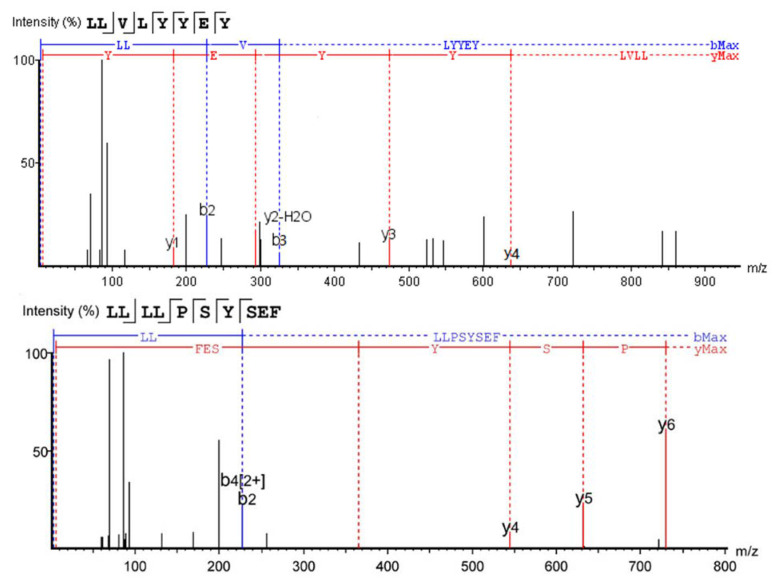
Mass spectra of two novel *α-glucosidase* inhibitory peptides.

**Figure 6 foods-12-00393-f006:**
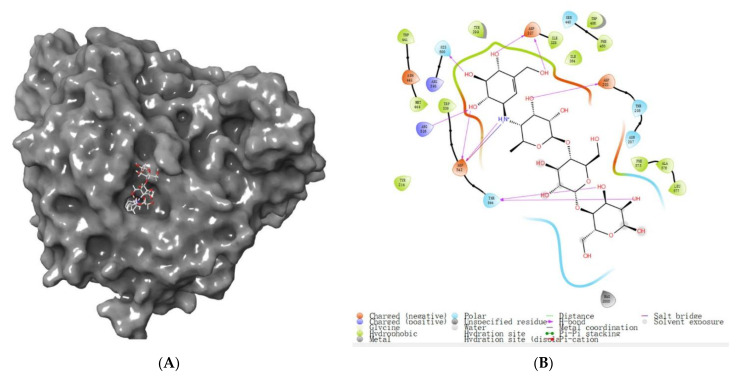
Binding pattern of *α-glucosidase* to acarbose. (**A**) Acarbose and *α-glucosidase* complex’s overall 3D structure. (**B**) 2D diagram of *α-glucosidase* and acarbose protein–ligand interaction.

**Figure 7 foods-12-00393-f007:**

Lineweaver-Burk diagram of *α-glucosidase* inhibition by (**A**) LLVLYYEY and (**B**) LLLLPYYSEF. Slope versus concentration graph (**C**) LLVLYYEY and (**D**) LLLLPSYSEF to calculate K_ic_. 1/Vmax versus concentration graph (**E**) LLVLYYEY and (**F**) LLLLPSYSEF to calculate K_iu_. All values are expressed as the mean of *α-glucosidase* inhibition. Testing is performed in triplicate.

**Figure 8 foods-12-00393-f008:**
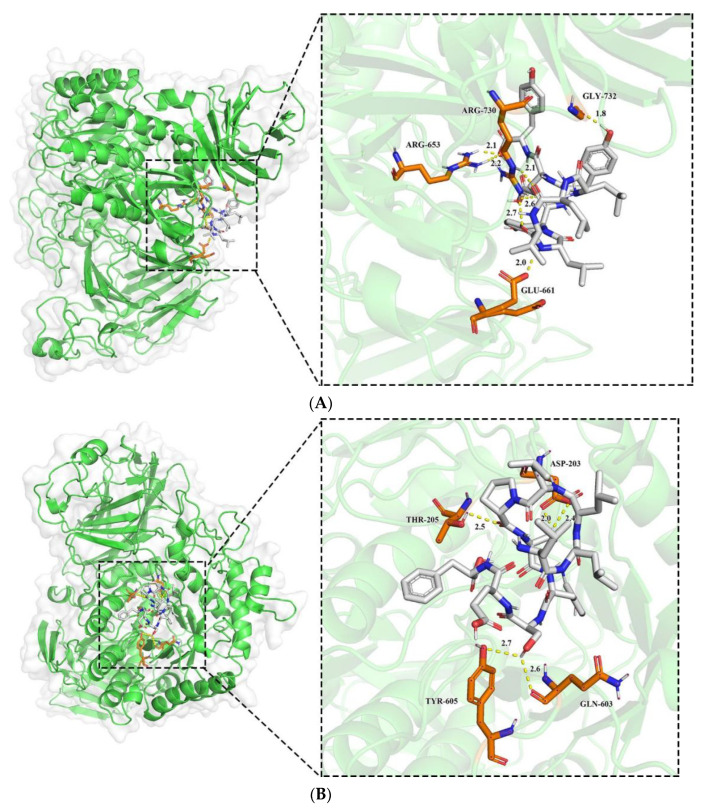
The 3D plot of peptide docking results with *α-glucosidase* (**A**): LLVLYYEY; (**B**): LLLLPSYSEF.

**Figure 9 foods-12-00393-f009:**
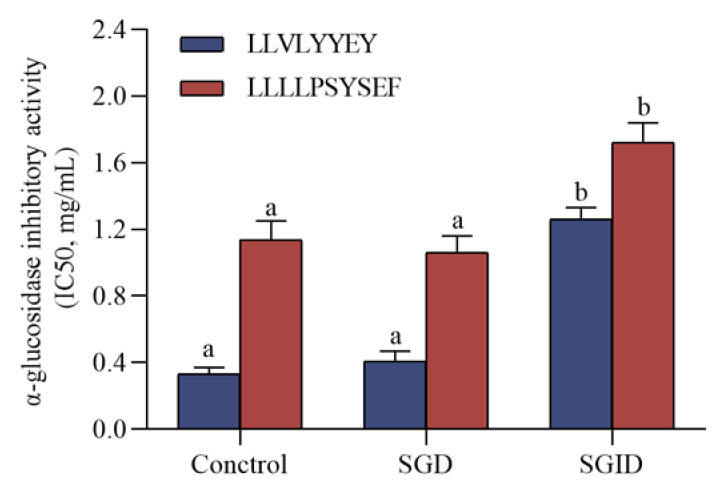
Simulated effects of gastrointestinal digestion on *α-glucosidase* inhibitory peptides. Different superscript letters in the same group indicate that they are significantly different (*p* < 0.05).

**Table 1 foods-12-00393-t001:** Regression model and ANOVA results based on *α-glucosidase inhibition activity*.

Source	Sum of Squares	df	Mean Square	F Value	*p*-Value	Significance
Model	992.59	14	70.90	21.37	<0.0001	**
A-Temperature	0.6120	1	0.6120	0.1845	0.6741	
B-pH	38.81	1	38.81	11.70	0.0041	**
C-Time	1.12	1	1.12	0.3383	0.5701	
D-Protease concentration	96.22	1	96.22	29.00	<0.0001	**
AB	0.7056	1	0.7056	0.2127	0.6518	
AC	0.6889	1	0.6889	0.2076	0.6556	
AD	0.3080	1	0.3080	0.0928	0.7651	
BC	0.0182	1	0.0182	0.0055	0.9420	
BD	0.1260	1	0.1260	0.0380	0.8483	
CD	0.5041	1	0.5041	0.1519	0.7025	
A^2^	24.75	1	24.75	7.46	0.0162	*
B^2^	249.52	1	249.52	75.21	<0.0001	**
C^2^	45.93	1	45.93	13.84	0.0023	**
D^2^	731.54	1	731.54	220.50	<0.0001	**
Residual	46.45	14	3.32			
Lack of Fit	43.18	10	4.32	5.29	0.0614	Not significance
Pure Error	3.27	4	0.8170			
Cor Total	1039.04	28				

Significance: ** means very significant, *p* < 0.01; * means significant, *p* < 0.05.

**Table 2 foods-12-00393-t002:** *α-glucosidase* inhibition activity (IC_50_ mg/mL) of the unfractionated CSCPH and the four fractions obtained via ultrafiltration.

Molecular Weight (kDa)	*α-glucosidase Inhibitory Activity* (IC_50,_ mg/mL)
CSCPH (Unfractionated)	8.442 ± 0.33 ^b^
CSCPH-I (MW <1 kDa)	59.450 ± 0.893 ^d^
CSCPH-II (1 kDa < MW < 3 kDa)	3.896 ± 0.148 ^a^
CSCPH-III (3 kDa < MW < 10 kDa)	62.440 ± 0.965 ^d^
CSCPH-IV (MW > 10 kDa)	15.800 ± 0.760 ^c^

All values are expressed as mean *α-glucosidase inhibitory activity* (IC_50_ values) ± Std. Deviation and tests were performed in triplicate. When superscript letters appear different in the same column, it indicates that they are significantly different (*p* < 0.05).

**Table 3 foods-12-00393-t003:** Amino acid compositions and *α-glucosidase inhibitory activity* of CSCP and the fractions obtained from different purification stages.

Amino Acids	CSCP (%)	CSCPH (%)	CSCPH-II (%)	CSCPH-II-4 (%)
Asp	7.64 ± 0.11 ^a^	8.22 ± 1.07 ^a^	7.92 ± 0.74 ^a^	6.79 ± 0.11 ^b^
Glu	19.78 ± 0.71 ^a^	21.18 ± 4.37 ^a^	16.47 ± 0.61 ^b^	10.42 ± 0.08 ^c^
Ser	3.98 ± 0.02 ^bc^	3.86 ± 0.48 ^c^	4.37 ± 0.32 ^a^	4.16 ± 0.13 ^ab^
Gly	3.94 ± 0.03 ^b^	3.93 ± 0.34 ^b^	4.84 ± 0.17 ^a^	4.91 ± 0.06 ^a^
His	1.48 ± 0.01 ^b^	1.54 ± 0.21 ^b^	1.72 ± 0.33 ^b^	4.03 ± 0.05 ^a^
Arg	2.42 ± 0.01 ^c^	2.47 ± 0.28 ^c^	3.81 ± 0.33 ^b^	4.41 ± 0.10 ^a^
Thr	4.92 ± 0.04 ^b^	4.79 ± 0.57 ^b^	6.19 ± 0.79 ^a^	5.70 ± 0.07 ^a^
Ala	15.81 ± 0.07 ^a^	16.48 ± 1.13 ^a^	13.79 ± 1.60 ^a^	7.90 ± 0.09 ^b^
Pro	10.89 ± 0.09 ^bc^	10.24 ± 1.31 ^c^	11.59 ± 1.02 ^b^	13.19 ± 0.13 ^a^
Tyr	2.55 ± 0.08 ^b^	2.53 ± 0.30 ^b^	2.58 ± 0.15 ^b^	6.67 ± 0.07 ^a^
Val	2.86 ± 0.02 ^b^	2.64 ± 0.32 ^b^	3.77 ± 0.41 ^a^	3.81 ± 0.04 ^a^
Met	2.53 ± 0.04 ^a^	2.40 ± 0.33 ^a^	2.50 ± 0.19 ^a^	0.81 ± 0.03 ^b^
Leu	1.46 ± 0.10 ^b^	1.22 ± 0.03 ^b^	1.47 ± 0.17 ^b^	4.66 ± 0.06 ^a^
Ile	3.23 ± 0.02 ^c^	3.20 ± 0.34 ^c^	3.90 ± 0.21 ^b^	4.21 ± 0.05 ^a^
Phe	7.01 ± 0.03 ^ab^	6.70 ± 0.78 ^b^	7.08 ± 0.57 ^ab^	7.43 ± 0.14 ^a^
Trp	3.49 ± 0.02 ^c^	3.31 ± 0.36 ^d^	3.97 ± 0.15 ^b^	7.41 ± 0.16 ^a^
Lys	6.00 ± 0.05 ^a^	5.28 ± 0.64 ^b^	4.02 ± 0.38 ^c^	3.48 ± 0.09 ^d^
EAA	31.51 ± 0.13 ^b^	29.58 ± 1.11 ^c^	32.87 ± 0.95 ^b^	37.50 ± 0.41 ^a^
HAA	47.30 ± 0.28 ^ab^	46.36 ± 1.50 ^b^	48.15 ± 1.61 ^ab^	49.47 ± 0.41 ^a^
BCAA	7.55 ± 0.06 ^c^	7.09 ± 0.27 ^d^	9.14 ± 0.08 ^b^	12.63 ± 0.33 ^a^
*α-glucosidase inhibitory* (IC_50_, mg/mL) activity (IC50, mg/mL)	nd	8.44 ± 0.33 ^a^	3.90 ± 0.15 ^b^	2.03 ± 0.09 ^c^

EAA, essential amino acid; HAA, hydrophobic amino acid; BCAA, branch amino acid; nd, no *α-glucosidase inhibitory activity* was detected. When superscript letters appear different in the same row, it indicates that they are significantly different (*p* < 0.05).

**Table 4 foods-12-00393-t004:** Screening peptides for in vitro *α-glucosidase inhibitory activity*.

Peptides Sequence	de Novo Score	Mass	m/z	XP Score (kcal/mol)	*α-glucosidase Inhibitory* *Activity* (IC_50_, mM)
LLVLYYEY	94	1074.56	538.287	−9.335	0.33
LLLLPSYSEF	92	1180.64	591.3289	−9.060	1.11
LCDQCPPHA	91	1096.44	549.229	−8.535	4.32
ATNPPCCQP	90	1043.42	522.7232	−8.868	>10
KDDFVEKR	99	1035.53	518.7742	−8.307	>10

## Data Availability

Data is contained within the article.

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
