# Peer review of "Optimization and Molecular Mechanism of Novel α-Glucosidase Inhibitory Peptides Derived from Camellia Seed Cake through Enzymatic Hydrolysis"

_foods, 2023, doi:10.3390/foods12020393_

Round 1

Reviewer 1 Report (Previous Reviewer 2)

the manuscript has already been revised carefully by the authors.

Author Response

感谢您对本文的认可

Reviewer 2 Report (Previous Reviewer 1)

The authors responded to reviewer’s comments and revised.

My comments on the revision and their response are as follows.

In general, it has been demonstrated that peptides are stable against endo type proteinase but easily digested by exopeptidase (J. Agric. Food Chem., 67 (43), 11948-11954, 2019). Most peptides larger than tripeptides are degraded by exopeptidases. Biological activity of peptides without considering stability in the digestive tract and content in target organs upon ingestion is difficult to be associated with in vivo response. The authors added simulated digestion. Please provide this system consisting of exopeptidases.

To the comment 2:

The authors just described that they used a similar procedure to the previous study. Reader cannot understand how to identify. The method (such as matching database of protein sequence) and name of database and name of used proteins should be provided.

To the comment 3

The authors described the principle of general identification method in their response and provided a revised spectrum. By seeing the revised spectrum in their response, I can understand this peptide sequence is LLVLYYEY. However, the data presented in the revised manuscript (Figure 5) were different patterns from that shown there. In Figure 5 upper, immonium ions, a2, b2, a3, and b6 were provided for LLVLYYEY. Readers still cannot understand why the authors identified them on the basis of the data in Figure 5 in the revised manuscript.

To comment 12

The authors provided amino acid composition data in Table 3.

Trp was determined. How did you release Trp from peptide fractions? It needs explanation, as it is destroyed by normal HCl hydrolysis.

Comment 13

The authors mentioned it was difficult to determine identified peptide content in the hydrolysate as so many peptides were present in hydrolysates. However, they detected these peptides in total ion scan mode and identified their sequences by product ion scan mode. In addition, they synthesized the peptides. It is much easier to determine the known peptide with MRM mode compared to detecting peptide by total ion scan mode in such a complex mixture.

Author Response

  • 请检查附件

This manuscript is a resubmission of an earlier submission. The following is a list of the peer review reports and author responses from that submission.

Round 1

Reviewer 1 Report

The authors prepared enzymatic hydrolysates of camellia seed cake protein using 4 different proteases. They demonstrated that the low molecular weight peptide fraction of protamex digest showed strong alpha-glucosidase inhibitory activity and identified many peptides in the active fraction using LC-MS/MS. Based on molecular docking and in vitro assays, two alpha-glucosidase inhibitory peptides, LLVLYYEY and LLLLPSYSEF, were identified. It looks interesting but I feel uneasy.

Before discussion of the content, the authors did not provide sufficient data and wrong expression.

No description for methods for peptide synthesis.

They identified peptides based on product ion scan using LC-MS/MS. These peptides consisted of Leu. But Leu and Ile cannot be distinguished by LC-MS. How did you identify as Leu? Please show the evidence.

In Figure 5, fragment ions of peptides were shown. How did you identified LLVLYYET based on only y1, b2, y3 ions? How did you estimate sequence VLY not LYV, YVL, YLV? If the authors identified them using protein sequences in the database, please show the parent protein.

L34 “diabetics is” may be “diabetics was”

L120: Sephadexed should be Sephadex

L223: “papain” No papain in Figure 1!

L366: “amino acid content” should be “amino acid composition”.

L373: “gel chromatography” should be “gel filtration chromatography”.

L374: “four components were identified”?? “four fractions were collected”?

L393; “the peptide sequence of CSCPH-II-4” may be “the sequence peptides in CSCPH-II-4”

L402 and L405: “Figure 5” must be “Figure 6”!!!

L419: “amino acid analysis” Where was the data? Amino acid analysis provides amino acid content or amino acid composition.

Main points

The authors identified two alpha-glucosidase inhibitors (I suspected its sequence based on the data in Figure 5). The authors used a molecular docking method to screen the alpha-glucosidase inhibitory peptide. Please show the contents of two peptides in the hydrolysate and discuss their contribution to alpha-glucosidase inhibitory activity of the digest. If the contents of identified peptides were so small, their contribution to the alpha-glucosidase inhibitory activity of the digest is low.

As mentioned in the text, stability of the identified peptides in the digestive tract is very important.

Reviewer 2 Report

the manuscript entitled Optimisation and Molecular mechanism of novel α-lucosidase inhibitory peptides Derived from Camellia seed Cake through Enzymatic Hydrolysis is quite interesting. some of comments and suggestions are addressed to improve the manuscript quality.

1. please makes the abbreviations/ nomenclatures. it will help readers to understand for example LLVLYYEY and LLLLPSYSEF

2. please add more explanation in introduction related to optimization

3. please add the objective in the end of introduction. please make it in the one paragraph

4. please add the section of chemical preparations

5. why do you choose the BBD instead of CCD. commonly, 4 parameters use the CCD instead of BBD.

6.  the proofread should be done to this manuscript, there are some of mistakes and typo.

7.the equation number of optimization model should be added and discuss in the manuscript

8.the sentence of "....which showed F-values of 21.37 and an extremely low p-value (p 0.0001)"?? P<0.0001/p>0.0001. please check the symbol all the manuscript.

9. please add the future perspective in the end of abstract

10. please revise the axis of figure 3(c)

Author Response

请看附件
